# Bi^3+^ and Eu^3+^ Activated Luminescent Behaviors in Non-Stoichiometric LaO_0.65_F_1.7_ Structure

**DOI:** 10.3390/ma13102326

**Published:** 2020-05-19

**Authors:** Sungjun Yang, Sangmoon Park

**Affiliations:** Division of Energy Convergence Engineering, Major in Energy & Applied Chemistry, Silla University, Busan 46958, Korea; qse7417@naver.com

**Keywords:** X-ray diffraction, phosphors, Bi^3+^, Eu^3+^ transitions, energy transfer

## Abstract

Optical materials composed of La_1-_*_p_*_-_*_q_*Bi*_p_*Eu*_q_*O_0.65_F_1.7_ (*p* = 0.001–0.05, *q* = 0–0.1) were prepared via a solid-state reaction using La(Bi,Eu)_2_O_3_ and NH_4_F precursors at 1050 °C for two hours. X-ray diffraction patterns of the phosphors were obtained permitting the calculation of unit-cell parameters. The two La^3+^ cation sites were clearly distinguished by exploiting the photoluminescence excitation and emission spectra through Bi^3+^ and Eu^3+^ transitions in the non-stoichiometric host lattice. Energy transfer from Bi^3+^ to Eu^3^^+^ upon excitation with 286 nm radiation and its mechanism in the Bi^3+^- and Eu^3+^-doped host structures is discussed. The desired Commission Internationale de l’Eclairage values, including emissions in blue-green, white, and red wavelength regions, were obtained from the Bi^3+^- and Eu^3+^-doped LaO_0.65_F_1.7_ phosphors.

## 1. Introduction

Ce^3+^-doped Y_3_Al_5_O_12_ (YAG) yellow phosphors are commonly used with blue light-emitting diodes (LEDs) to create white light sources [1,2,3,4,5]. The Ce^3+^ ions emit in the blue to yellow wavelength regions assigned by 5*d^1^* to 4*f^1^* transitions when excited by ultraviolet (UV) to visible radiation in various host lattices [6,7,8,9]. The Ce^3+^ ion, as a donor, enables efficient energy transfer, improving the emission from acceptors, such as Tb^3+^ or Mn^2+^ ions in the host structures [10,11,12,13,14,15,16,17,18]. The Bi^3+^ ion is an active luminescent center emitting blue to green light assigned to 6*s*^1^6*p*^1^ to 6*s*^2^ transitions when excited by UV to near UV wavelength regions in host lattices [19,20,21]. The energy levels of the Bi^3+^ 6*s*^2^–6*s*^1^6*p*^1^ transitions consist of ^1^S_0_ and the triplet ^3^P_J_ (J = 0, 1, or 2) and singlet ^1^P_1_ states. The ^1^S_0_ to ^3^P_1_, ^1^P_1_ transitions occur via spin-orbital coupling [19,20,21]. The states of the ^1^S_0_ to ^3^P_0_ and ^3^P_2_ transitions are forbidden [19,20,21]. Like Ce^3+^ ions, Bi^3+^ ions act as sensitizers to enhance the anticipated emission light from acceptors, such as Eu^3+^ or Tb^3+^ ions in host structures, by facilitating efficient energy transfer [22,23,24,25,26].

The up-conversion properties of Er^3+^- and Yb^3+^-doped LaO_0.65_F_1.7_ compounds were exploited under 980 nm diode laser excitation in a previous study [27]. This non-stoichiometric LaO_0.65_F_1.7_ host comprises alternating stacked LaO_2_F_7_ and LaO_3_F_7_ layers along the *c* axis with tetragonal space group *P4/nmm*, as shown in Figure 1 [27,28]. The 9- and 10-coordinated La^3+^ sites in the LaF(1)_3_F(2)_2_O_2_F(3)_2_ and LaF(1)_4_F(2)O_3_F(3)_2_ polyhedrons are located in accordance with the La(F(1)_0.86_V_0.14_)(F(2)_0.35_O_0.65_)(F(3)_0.49_) lattice of the LaO_0.65_F_1.7_ host structure [27,28]. Notably, the nine-fold LaF(1)_3_VF(2)_2_O_2_F(3)_2_ polyhedron contains a vacancy (V) associated with the F(1) anion.

In this study, Bi^3+^ and Eu^3+^ were substituted into LaO_0.65_F_1.7_ compounds that were synthesized by a solid-state method using NH_4_F flux in air. The unit-cell parameters of the phosphors were calculated. The excitation and emission luminescence spectra of the La_1-*p*-*q*_Bi*_p_*E u*_q_*O_0.65_F_1.7_ (*p* = 0.001–0.05, *q* = 0–0.1) phosphors were investigated with respect to the site dependency of Bi^3+^ and Eu^3+^ ions in the host structure. The energy transfer mechanism from Bi^3+^ to Eu^3+^ in the phosphors was explored. Commission Internationale de l’Eclairage (CIE) chromaticity coordinates of the phosphors were obtained.

## 2. Materials and Methods

Phosphors of La_1-*p*-*q*_Bi*_p_*Eu*_q_*O_0.65_F_1.7_ (*p* = 0.005–0.05, *q* = 0–0.1) were prepared by heating the appropriate amounts of La_2_O_3_ (Alfa 99.9%), Bi_2_O_3_ (Alfa 99.99%), Eu_2_O_3_ (Alfa 99.9%), and NH_4_F (Alfa 99%). Powdered samples with 1:2 molar ratios of La(Bi,Eu)O_3/2_ and NH_4_F were used to prepare nonstoichiometric LaO_0.65_F_1.7_:Bi^3+^, Eu^3+^. The precursors were mixed with an agate mortar and pestle and subsequently heated at 1050 °C for 2 h in air [27]. The La_2_O_3_ precursor was pre-heated at 700 °C for 3 h to remove hydroxide in the sample. Phase identification of the phosphors was performed using a Shimadzu XRD-6000 powder diffractometer (Cu-Kα radiation, Shimadzu CO., Kyoto, Japan). The Rietveld refinement program Rietica was used for the unit-cell parameter calculations. UV spectroscopy of the excitation and emission spectra of the phosphors was measured using spectrofluorometers (Sinco Fluoromate FS-2, Sinco CO., Seoul, Korea).

## 3. Results and Discussion

The crystallographic phase of the La_1-*p*-*q*_Bi*_p_*Eu*_q_*O_0.65_F_1.7_ (*p* = 0.001–0.05, *q* = 0–0.1) powders was identified using powder X-ray diffraction (XRD) patterns. The calculated XRD pattern of the tetragonal LaO_0.65_F_1.7_ (ICSD 40371) structure is shown in Figure 2A. Figure 2B–F show the XRD patterns of non-stoichiometric La_1-*p*-*q*_Bi*_p_*Eu*_q_*O_0.65_F_1.7_ phosphors (*p* = 0.01 and *q* = 0, *p* = 0.05 and *q* = 0, *p* = 0 and *q* = 0.05, *p* = 0 and *q* = 0.1, and *p* = 0.01 and *q* = 0.1, respectively), synthesized by the mixing of ½La(Bi,Eu)_2_O_3_ and NH_4_F at 1050 °C in air. The XRD patterns of the obtained phosphors in Figure 2B–F show a single-phase structure without any noticeable impurities indexed to a tetragonal unit cell. The unit cells of La_0.99_Bi_0.01_O_0.65_F_1.7_, La_0.95_Bi_0.05_O_0.65_F_1.7_, La_0.95_Eu_0.05_O_0.65_F_1.7_, La_0.9_Eu_0.1_O_0.65_F_1.7_, and La_0.89_Bi_0.01_Eu_0.1_O_0.65_F_1.7_ phosphors were calculated to be *a* = 4.0934 (1) Å and *c* = 5.8336 (2) Å, *a* = 4.1018 (2) Å and *c* = 5.8315 (2) Å, *a* = 4.0833 (2) Å and *c* = 5.8162 (4) Å, *a* = 4.0788 (3) Å and *c* = 5.8095 (5) Å, and *a* = 4.0993(3) Å and *c* = 5.7712(6) Å, respectively, using the Rietveld refinement. The unit-cell parameters, including the cell volumes of the phosphors, are summarized in Table 1. The Bi^3+^ and Eu^3+^ ions, under these conditions, occupy 9- and 10-coordinated La^3+^ sites (LaF(1)_3_F(2)_2_O_2_F(3)_2_ and LaF(1)_4_F(2)O_3_F(3)_2_) in the non-stoichiometric LaO_0.65_F_1.7_ structure, as shown in Figure 1 [27,28]. The single La^3+^ site comprises 56% 9-fold and 44% 10-fold polyhedrons in the LaO_0.65_F_1.7_ lattice based on the La(F(1)_0.86_V_0.14_)(F(2)_0.35_O_0.65_)(F(3)_0.49_) formula. The 9- and 10- coordinated LaO_2_F_7_ and LaO_3_F_7_ polyhedrons in the non-stoichiometric unit cell are arrayed along the *c*-axis, as shown in Figure 1. When Bi^3+^ ions (*r* = 1.17 Å for 8 coordination number (CN)) were substituted for La^3+^ ions (*r* = 1.16 Å for 8 CN) in the LaO_0.65_F_1.7_ host lattice, gradual shifts in the positions of the various Bragg reflections to lower angles with unit-cell expansion were observed, as shown in Figure 2B,C. When Eu^3+^ ions (*r* = 1.066 Å for 8 CN) were substituted for La^3+^ ions in the host lattice, gradual shifts in the positions of the various Bragg reflections to higher angles with unit-cell contraction were observed, as shown in Figure 2D,E. When the Bi^3+^ ions were doped in the La_0.9_Eu_0.1_O_0.65_F_1.7_ phosphors, no further shift to higher angles was observed in the La_0.89_Bi_0.01_Eu_0.1_O_0.65_F_1.7_ phosphors, as shown in Figure 2F.

Figure 3aA–E show the photoluminescence (PL), excitation (EX), and emission (EM) spectra of the Bi-doped La_1-*p*_Bi*_p_*O_0.65_F_1.7_ phosphors (*p* = 0.001, 0.005, 0.01, 0.025, and 0.05, respectively). The excitation band centered near 278 and 286 nm in the La_0.99_Bi_0.01_O_0.65_F_1.7_ PL spectra is attributed to the ^1^S_0_ → ^3^P_1_ transition of Bi^3+^ ions because the ^1^S_0_ → ^3^P_0_ and ^1^S_0_ → ^3^P_2_ transitions are forbidden from ground ^1^S_0_ [19,20,21,22,23,24,25,26]. The blue emission spectra of the LaO_0.65_F_1.7_:Bi^3+^ phosphors revealed a broadband range from 350 to 650 nm, centered at approximately 497 nm, which is attributed to the intense ^3^P_1_ → ^1^S_0_ transitions of the Bi^3+^ ions, as shown in Figure 3a. When the Bi^3+^ concentration in the host lattice was 1 mol %, the maximum emission intensity of the obtained phosphors was observed at the excitation wavelength of 278 nm, as shown in Figure 3aC. After the Bi^3+^ concentration was increased 2.5 mol % in the phosphors, the centered excitation peak shifted to a higher wavelength region from 278 to 286 nm, as shown in Figure 3aD,E. Thus, as the Bi^3+^ content in the LaO_0.65_F_1.7_ host lattice was increased and the excitation center of the ^1^S_0_ → ^3^P_1_ transition of Bi^3+^ ions underwent a shift to a longer wavelength. The La^3+^ ion is coordinated by seven F^−^ and two O^2−^ anions (LaF(1)_3_F(2)_2_O_2_F(3)_2_), or seven F^−^ and three O^2−^ anions (LaF(1)_4_F(2)O_3_F(3)_2_) in the LaO_0.65_F_1.7_ host structure [27,28]. As depicted in Figure 1, there was a vacancy associated with the F(1) anion in the LaF(1)_3_F(2)_2_O_2_F(3)_2_ polyhedron. Based on the ratios of oxygen and fluoride to lanthanum, the LaF(1)_3_F(2)_2_O_2_F(3)_2_ polyhedron had a lower oxygen ion covalency than LaF(1)_4_F(2)O_3_F(3)_2_ polyhedrons in the structure. This observation indicated that Bi^3+^ ions are preferentially substituted in the nine-fold La site and subsequently doped into the 10-fold La site in the host structure. Figure 3b shows the excitation and emission PL spectra of the La_0.95_Eu_0.05_O_0.65_F_1.7_ phosphors. The charge-transfer bands (CTBs) and the *f–f* transitions of the Eu^3+^ activator in the host lattice were observed at 220–350 and 350–540 nm, respectively. Two CTBs centered at 290 and 320 nm were found in the excitation spectra because there were two La^3+^ sites associated with the LaF(1)_3_F(2)_2_O_2_F(3)_2_ and LaF(1)_4_F(2)O_3_F(3)_2_ polyhedrons in the host structure. When Eu^3+^ ions were doped in the nine-coordinated La^3+^ site of the LaF(1)_3_F(2)_2_O_2_F(3)_2_ polyhedron, the center of the Eu^3+^ CTB transitions occurred at 290 nm. Additional energy was required to excite an electron from the Eu^3+^ ions in seven F^−^ and two O^2−^ containing lattices, compared to seven F^−^ and three O^2−^ polyhedrons.

The Eu^3+^ transitions of the emission spectra in the La_0.95_Eu_0.05_O_0.65_F_1.7_ phosphors exhibited both the ^5^D_0_–^7^F_1_ magnetic dipole and the ^5^D_0_–^7^F_2_ electric-dipole transitions, centered at 592 and 610 nm, respectively [29,30]. When the Eu^3+^ ions were substituted in no inversion site of the nine-coordinated polyhedron in the host lattice, the ^5^D_0_–^7^F_2_ transition dominates. When the Eu^3+^activators were doped into symmetric inversion site of the 10-fold polyhedron, the ^5^D_0_–^7^F_1_ transition dominates. Figure 3c shows the excitation spectra of the La_0.95_Eu_0.05_O_0.65_F_1.7_ (EX_EM=610nm_ and EX_EM=592nm_) and the emission spectrum of La_0.99_Bi_0.01_O_0.65_F_1.7_ (EM_EX=286nm_) phosphors. The efficiency of energy transfer from Bi^3+^ to Eu^3+^ was estimated by the spectral overlap between the excitation of the Eu^3+^ transition and the emission band of Bi^3+^ ions in the host lattice [31]. The excitation spectrum of the La_0.95_Eu_0.05_O_0.65_F_1.7_ (EX_EM=610nm_) phosphor and the emission spectrum of the La_0.99_Bi_0.01_O_0.65_F_1.7_ (EM_EX=286nm_) phosphor exhibited considerable overlap, as shown in the top of Figure 3c. This indicated that effective energy transfer from Bi^3+^ to Eu^3+^ ions occurs in the nine-coordinated La^3+^ site of the LaF(1)_3_F(2)_2_O_2_F(3)_2_ polyhedron in the LaO_0.65_F_1.7_ host structure. The individual transitions of Bi^3+^ and Eu^3+^ ions with the energy transfer from Bi^3+^ to Eu^3+^ ions in the phosphors can simultaneously occur under approximately 290 nm excitation wavelength. However, the energy transfer was effectively observed rather than the individual transitions because the integrated emission intensity of Eu^3+^ transition was enhanced by approximately 91% from La_0.95_Eu_0.05_O_0.65_F_1.7_ to La_0.94_Bi_0.05_O_0.65_F_1.7_ phosphors (Appendix A). The blue-green emission of the La_1-*p*_Bi*_p_*O_0.65_F_1.7_ phosphors centered at 497 nm reached a maximum intensity for a Bi^3+^ content (*p* = 0.01), as shown in Figure 3a. After increasing the Bi^3+^ content, concentration quenching of the relative emission intensity was observed. The increase in the Bi^3+^ content of the phosphors enhanced energy transfer up to some critical value, whereas after this value was reached subsequent increase of Bi^3+^ levels decreased the emission intensity by reducing the critical distance between the Bi^3+^ ions. This resulted in non-radiative energy transfer between Bi^3+^ ions from the electric multipole interactions. The critical distance (*R*_c_) is expressed by the following formula:*R*_c_ = 2[3*V*/4π*m*_c_*N*]^1/3^(1)
where *V* is the volume of the La_0.99_Bi_0.01_O_0.65_F_1.7_ unit cell, *N* is the number of available La^3+^ sites for the dopant in the unit cell, *m*_c_ is the critical concentration of Bi^3+^, and *R*_c_ is the critical distance for energy transfer [10,22,23,24,32]. When *N* and *V* are 1 and 97.75 Å^3^, respectively, for La_0.99_Bi_0.01_O_0.65_F_1.7_, *R*_c_ (*m*_c_ = 0.01) is 26.53 Å. The energy transfer mechanism designated an electric multipole interaction because the critical distance is greater than 5 Å. Figure 4a shows the emission spectra of La_0.99-*q*_Bi_0.01_Eu*_q_*O_0.65_F_1.7_ (*q* = 0–0.1) phosphors under 286 nm excitation. Co-doping of Eu^3+^ into the Bi^3+^-doped LaO_0.65_F_1.7_ host structure allowed effective energy transfer from Bi^3+^ to Eu^3+^ under excitation at 286 nm. The energy transfer from Bi^3+^ to Eu^3+^ acted as a sensitizer and an activator, respectively, in the La_0.99-*q*_Bi_0.01_Eu*_q_*O_0.65_F_1.7_ (*q* = 0–0.1) phosphors, which was activated through the absorption from Bi^3+^ transitions. The energy transfer efficiency (η_T_) was evaluated using the following formula:η_T_ = 1 − *I*_S_/*I*_SO_(2)
where *I*_S_ and *I*_SO_ are the luminescence intensities of the Bi^3+^ sensitizer in the presence and absence of a Eu^3+^ activator, respectively [10,22,23,24,32]. The emission of Eu^3+^ transitions was maximized when the Eu^3+^ content in the La_0.99−*q*_Bi_0.01_Eu*_q_*O_0.65_F_1.7_ (*q* = 0–0.1) phosphors was *q* = 0.05. The energy transfer mechanism could be represented by linear plots of *I*_SO_/*I*_S_ versus C_Bi-Eu_^α/3^, where C_Bi-Eu_ is the concentration of Bi^3+^ and Eu^3+^ ions, with α = 6, 8, or 10, corresponding to dipole–dipole, dipole–quadrupole, and quadrupole–quadrupole interactions, respectively, in accordance with Dexter theory [10,22,23,24,32]. In Figure 4b, when α = 6, 8, and 10, the linear plots showed energy transfer from the Bi^3+^ to Eu^3+^ ions with *R*^2^ = 0.9635, 0.9894, and 0.9982 in the La_0.94_Bi_0.01_Eu_0.05_O_0.65_F_1.7_ phosphors, respectively. As the value of α is 10, a closer linear plot is determined for the phosphor, the quadrupole–quadrupole interaction was involved in the energy transfer mechanism of the La_0.94_Bi_0.01_Eu_0.05_O_0.65_F_1.7_ phosphors. The efficiency of the energy transfer from Bi^3+^ to Eu^3+^ in La_0.94_Bi_0.01_Eu_0.05_O_0.65_F_1.7_ (EX = 286 nm) phosphors is shown in Figure 4c. The efficiency was gradually enhanced from 23% to 97% as the Eu^3+^ content in the phosphors increased from *q* = 0.01 to 0.1.

As shown in Figure 5a, the chromaticity coordinates, x and y, are in accordance with the desired CIE (Commission Internationale de l’Eclairage) values from the blue-green to white and red wavelength regions for La_0.99-*q*_Bi_0.01_Eu*_q_*O_0.65_F_1.7_ (*q* = 0–0.1) phosphors (*EX* = 286 nm). The CIE values are summarized in the inset of Figure 5a, along with the values obtained for the phosphors. The CIE coordinates near the blue-green, white, orange, and red regions of the CIE diagram from the phosphors were observed to be *x* = 0.240 and *y* = 0.334, *x* = 0.328 and *y* = 0.348, *x* = 0.466 and *y* = 0.354, and *x* = 0.591 and *y* = 0.353, for values of *q* = 0, 0.02, 0.05, and 0.1, respectively. When the concentration of Eu^3+^ ions in the La_0.99-*q*_Bi_0.01_Eu*_q_*O_0.65_F_1.7_ phosphors increased from *q* = 0 to 0.02 and 0.1, the emission colors exhibited a significant shift from blue-green to white, and red emission regions, respectively. These tunable emission lights are appropriate for a high color-rendering index to apply phosphor converted UV-LEDs. This indicates that there was effective energy transfer from Bi^3+^ to Eu^3+^ in the La_0.99-*q*_Bi_0.01_Eu*_q_*O_0.65_F_1.7_ phosphors. Emission of the La_0.99-*q*_Bi_0.01_Eu*_q_*O_0.65_F_1.7_ (*q* = 0–0.1) phosphors under 254, 312, and 365 nm hand-lamp excitation was exhibited blue-green, white, orange, and red colors, as shown in Figure 5b.

## 4. Conclusions

Non-stoichiometric tetragonal La_1-*p*-*q*_Bi*_p_*Eu*_q_*O_0.65_F_1.7_ (*p* = 0.001–0.05, *q* = 0–0.1) phosphors were prepared via a solid-state method using a heat treatment at 1050 °C for two hours using NH_4_F flux. The site dependency of the Bi^3+^ and Eu^3+^ ions in the LaF(1)_3_F(2)_2_O_2_F(3)_2_ and LaF(1)_4_F(2)O_3_F(3)_2_ polyhedrons of the host structure was analyzed using the PL spectra of the phosphors. The maximum luminescence intensity of the blue-green La_1-*p*_Bi*_p_*O_0.65_F_1.7_ phosphors was obtained when *p* = 0.01. The critical distance (*R*c) value for the La_0.99_Bi_0.01_O_0.65_F_1.7_ phosphor was determined to be 26.53 Å. As the Eu^3+^ concentration was increased in La_0.99-*q*_Bi_0.01_Eu*_q_*O_0.65_F_1.7_ (*q* = 0–0.1) phosphors under 286 nm excitation, an efficient energy transfer from Bi^3+^ to Eu^3+^ occurred, involving quadrupole–quadrupole interactions in the phosphors. The CIE coordinate values attributed to the emissions from blue-green, white, and red for La_0.99-*q*_Bi_0.01_Eu*_q_*O_0.65_F_1.7_ (*q* = 0–0.1) phosphors were successfully obtained.

## Figures and Tables

**Figure 1 materials-13-02326-f001:**
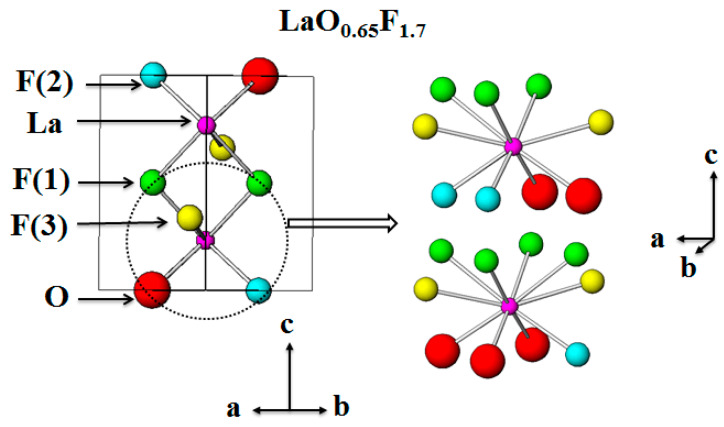
The structure of tetragonal LaO_0.65_F_1.7_ host lattice.

**Figure 2 materials-13-02326-f002:**
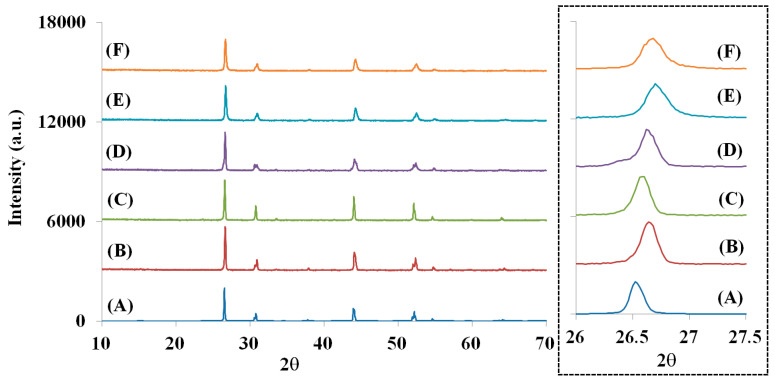
The calculated XRD patterns of (A) LaO_0.65_F_1.7_ (ICSD 40371) and the obtained XRD patterns of La_1-*p*-*q*_Bi*_p_*Eu*_q_*O_0.65_F_1.7_ phosphors (B) *p* = 0.01 and *q* = 0, (C) *p* = 0.05 and *q* = 0, (D) *p* = 0 and *q* = 0.05, (E) *p* = 0 and *q* = 0.1, and (F) *p* = 0.01 and *q* = 0.1.

**Figure 3 materials-13-02326-f003:**
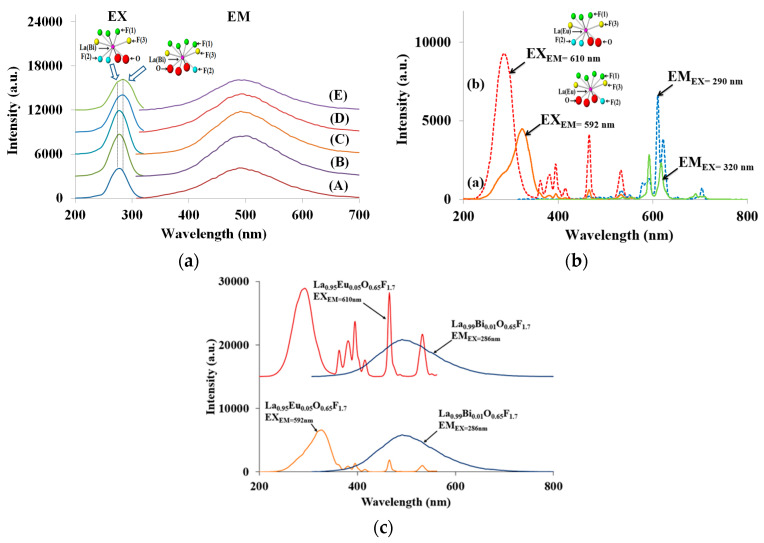
PL excitation and emission spectra of (**a**) La_1-*p*_Bi*_p_*O_0.65_F_1.7_ phosphors (A) *p* = 0.001, (B) 0.005, (C) 0.01, (D) 0.025, and (E) 0.05; and (**b**) La_0.95_Eu_0.05_O_0.65_F_1.7_ phosphors; and (**c**) the excitation spectra of the La_0.95_Eu_0.05_O_0.65_F_1.7_ and the emission spectrum of La_0.99_Bi_0.01_O_0.65_F_1.7_.

**Figure 4 materials-13-02326-f004:**
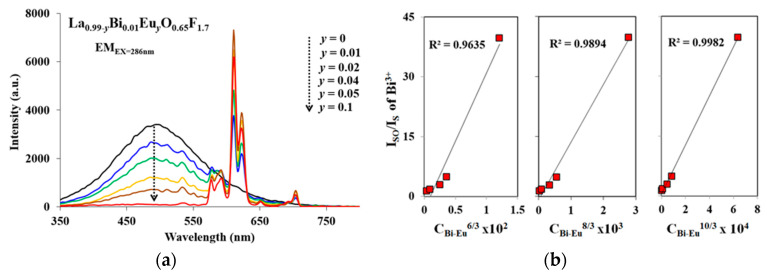
(**a**) The emission spectra of La_0.99-*q*_Bi_0.01_Eu*_q_*O_0.65_F_1.7_ (*q* = 0–0.1) phosphors under 286 nm excitation, (**b**) the plot of I_SO_/I_S_ versus C_Bi-Eu_^α/3^ (α = 6, 8, 10), and (**c**) energy transfer efficiency from Bi^3+^ to Eu^3+^ in the phosphors.

**Figure 5 materials-13-02326-f005:**
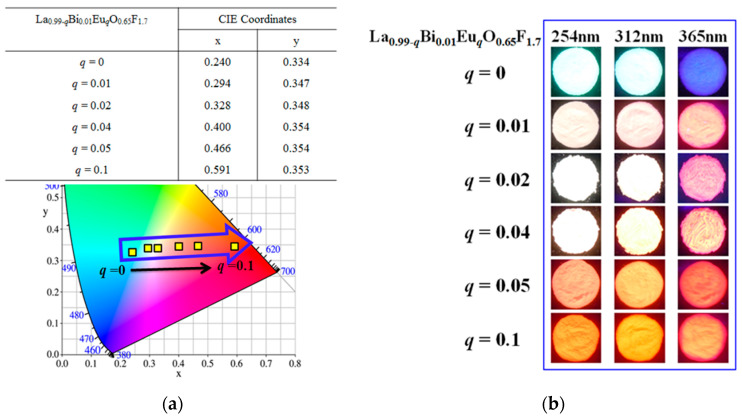
(**a**) The chromaticity coordinates with the desired CIE values of La_0.99-*q*_Bi_0.01_Eu*_q_*O_0.65_F_1.7_ (*q* = 0–0.1) phosphors (*EX* = 286 nm) and (**b**) photographs of the emission light from blue-green to white, orange, and red colors in the La_0.99−*q*_Bi_0.01_Eu*_q_*O_0.65_F_1.7_ phosphors under 254, 312, and 365 nm hand-lamps.

**Table 1 materials-13-02326-t001:** The unit-cell parameters with the cell volumes of the La_0.99_Bi_0.01_O_0.65_F_1.7_, La_0.95_Bi_0.05_O_0.65_F_1.7_, La_0.95_Eu_0.05_O_0.65_F_1.7_, La_0.9_Eu_0.1_O_0.65_F_1.7_, and La_0.89_Bi_0.01_Eu_0.1_O_0.65_F_1.7_ phosphors.

Phosphors	*a* (Å)	*c* (Å)	*V* (Å^3^)	*R_p_*
La_0.99_Bi_0.01_O_0.65_F_1.7_	4.0934 (1)	5.8336 (2)	97.75 (1)	9.11
La_0.95_Bi_0.05_O_0.65_F_1.7_	4.1018 (2)	5.8315 (2)	98.11 (1)	9.98
La_0.95_Eu_0.05_O_0.65_F_1.7_	4.0833 (2)	5.8162 (4)	96.98 (1)	9.46
La_0.9_Eu_0.1_O_0.65_F_1.7_	4.0788 (3)	5.8095 (5)	96.65 (1)	8.95
La_0.89_Bi_0.01_Eu_0.1_O_0.65_F_1.7_	4.0993 (3)	5.7712 (6)	96.98 (1)	9.62

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
