# Peer review of "Bi^3+^ and Eu^3+^ Activated Luminescent Behaviors in Non-Stoichiometric LaO_0.65_F_1.7_ Structure"

_materials, 2020, doi:10.3390/ma13102326_

Round 1

Reviewer 1 Report

Park and co-workers report the solid state synthesis of Bi and Eu doped LaO0.65 F1.7 phosphors and evaluated their photoluminescence properties. The manuscript is well written and of particular importance for the audience of Materials.

The reviewer has some questions :

  • How the authors explain that when the concentration of Bi3+ increase to 2.5 mol% the emission spectrum shift to higher wavelength ?
  • For the experiments with Bi and Eu doped LaO65 F1.7 phosphors the authors chose the 0.1mol % Bi concentration and varied the amount of Eu. Why do they chose this concentration ?
  • Did they tried with the 2.5mol % Bi concentration ?
  • If yes what does it changed ?
  • Do the authors have an idea of how long their phosphors will emit light?

Author Response

Park and co-workers report the solid state synthesis of Bi and Eu doped LaO0.65 F1.7 phosphors and evaluated their photoluminescence properties. The manuscript is well written and of particular importance for the audience of Materials.

The reviewer has some questions :

  • How the authors explain that when the concentration of Bi3+ increase to 2.5 mol% the emission spectrum shift to higher wavelength ? 
  • If it is the case of emission spectra as the review has mentioned, we could calculate the fitting of two components via Gaussian deconvolution for the Bi3+ activator centers, the preferred Bi3+ ion sites in the host structure can be determined. In this draft, we have studied in the use of the center of excitation spectra.

  • For the experiments with Bi and Eu doped LaO65 F1.7 phosphors the authors chose the 0.1mol % Bi concentration and varied the amount of Eu. Why do they chose this concentration ? Did they tried with the 2.5mol % Bi concentration ? If yes what does it changed ?
  •  When the blue-green emission of the La1-pBipO0.65F1.7 (p = 0.01) phosphors reached a maximum intensity, the energy transfer can fully arise from Bi3+ to Eu3+ ions in the La0.99-qBi0.01EuqO0.65F1.7 (q = 0 – 0.1) phosphors. After the increase of the concentration of Bi3+ ions like 2.5 mol%, concentration quenching normally occurs. As we tested the 2.5mol%Bi-Eu phosphors, the intensity of the emission was decreased but the CIE coordinates were similar with the coordinates of 1 mol%Bi-Eu phosphors as shown in below.  (Figures were added.)

  • Do the authors have an idea of how long their phosphors will emit light?
  • When the emitting light was tested under UV-handhold lamp, the emission light was quickly gone after the lamp was turned off.

Reviewer 2 Report

The work Bi3+ and Eu3+ activated luminescent behaviors in non-stoichiometric  LaO0.65F1.7 structure by Sungjun Yang and Sangmoon Park concerns the energy transfer between Bi3+ and Eu3+ in the LiOF host lattice. The work is well written and organized. The structural analysis raises no objections, but I have a few comments on the spectroscopic part. The authors are asked to pay attention to the following issues:

  1. To better document Bi3+ ® Eu3+ energy transfer, the excitation spectrum of Eu3+ emission in the Bi3+ and Eu3+ doped phosphor should be included. The Bi3+ and Eu3+ absorption bands overlap and it is not known if only the energy transfer occurs or the simultaneous excitation of both ions.
  2. We know nothing about the efficiency of this phosphor. It is worth determining the quantum efficiency of its emission and comparing this value to similar phosphors, for example, LiOF: Eu3 +.

Author Response

The work Bi3+ and Eu3+ activated luminescent behaviors in non-stoichiometric  LaO0.65F1.7 structure by Sungjun Yang and Sangmoon Park concerns the energy transfer between Bi3+ and Eu3+ in the LiOF host lattice. The work is well written and organized. The structural analysis raises no objections, but I have a few comments on the spectroscopic part. The authors are asked to pay attention to the following issues:

  1. To better document Bi3+ Eu3+ energy transfer, the excitation spectrum of Eu3+ emission in the Bi3+ and Eu3+ doped phosphor should be included. The Bi3+ and Eu3+ absorption bands overlap and it is not known if only the energy transfer occurs or the simultaneous excitation of both ions.

Text and figure were added.

The individual transitions of Bi3+ and Eu3+ ions with the energy transfer from Bi3+ to Eu3+ ions in the phosphors can simultaneously occur under approximately 290 nm excitation wavelength; however, the energy transfer was effectively observed rather than the individual transitions because the integrated emission intensity of Eu3+ transition was enhanced approximately 91 % from La0.95Eu0.05O0.65F1.7 to La0.94Bi0.05O0.65F1.7 phosphors (S1).

S1. The integrated emission intensities of La0.95Eu0.05O0.65F1.7 and La0.94Bi0.05O0.65F1.7 phosphors.

  1. We know nothing about the efficiency of this phosphor. It is worth determining the quantum efficiency of its emission and comparing this value to similar phosphors, for example, LiOF: Eu3 +.

As the commercial LaOF:Eu was not known as far as I know, In our present study, the integrated emission intensity of the La0.95Eu0.05OF and La0.95Eu0.05O0.65F1.7 was compared. The intensity was enhanced approximately 85 % from La0.95Eu0.05OF to La0.95Eu0.05O0.65F1.7 phosphors (Please check the attached file), which can be estimated as a quantum efficiency, which was not shown in this draft.

Reviewer 3 Report

The subject is suitable for aims of journal. Authors presents a proposal of new materials phosphors but is more a structural research.The elements proposed in research are favored to be phosphors. 

The procedure to obtain seems to be extremely simple, and just using XRD data and UV-vis results it is constructed a manuscript about new materials.

How many samples were obtained?

Nomenclature of NH4F?

In my point of view at least of two methods could be an advantage in the quality of results.

There were included discussions about Ce+3, but is a difference between it and more stable stare of Ce +4, or this similarity could be for another ions. XPS analysis to confirm state of ion is necessary.

For me is unclear presented the applications of these materials, behavior of them, in different media, utility of studies.

 Stability of new materials, no destruction, changing in state of ions and characteristics? Solubility? and applicability?

 There are limited number of figures.

As a conclusion, the manuscript has be revised and more improved.

 It is more structural research than focused on more properties/ behavior  and applications, information more useful for readers.

Author Response

The subject is suitable for aims of journal. Authors presents a proposal of new materials phosphors but is more a structural research. The elements proposed in research are favored to be phosphors. 

The procedure to obtain seems to be extremely simple, and just using XRD data and UV-vis results it is constructed a manuscript about new materials.

How many samples were obtained? More than 100 samples were prepared.

Nomenclature of NH4F? It is ammonium fluoride.

In my point of view at least of two methods could be an advantage in the quality of results.

There were included discussions about Ce+3, but is a difference between it and more stable stare of Ce +4, or this similarity could be for another ions. XPS analysis to confirm state of ion is necessary.

We have prepared the optical materials composed of La1-p-qBipEuqO0.65F1.7 (p = 0.001 – 0.05, q = 0 – 0.1). Ce-doped samples were not studied in this draft; however, as far as I know the Ce4+ ions can be usually reduced to Ce3+ ions under reduction atmosphere in the synthetic process. When the Ce3+ ions was obtained, the emission can be also observed.

For me is unclear presented the applications of these materials, behavior of them, in different media, utility of studies. Stability of new materials, no destruction, changing in state of ions and characteristics? Solubility? and applicability?  There are limited number of figures.

As a conclusion, the manuscript has be revised and more improved.  It is more structural research than focused on more properties/ behavior and applications, information more useful for readers.

The solid-solution of the La1-xEuxO0.65F1.7 phosphors is currently studying in my lab. As you know, the stability of the phosphors is not excellent because the oxyfluorides can be oxidized in air and oxyfluoride phosphors are very weak in moisture as well.

Figure and text were added.

The individual transitions of Bi3+ and Eu3+ ions with the energy transfer from Bi3+ to Eu3+ ions in the phosphors can simultaneously occur under approximately 290 nm excitation wavelength; however, the energy transfer was effectively observed rather than the individual transitions because the integrated emission intensity of Eu3+ transition was enhanced approximately 91 % from La0.95Eu0.05O0.65F1.7 to La0.94Bi0.05O0.65F1.7 phosphors (S1).

The figure (S1) of the integrated emission intensities of La0.95Eu0.05O0.65F1.7 and La0.94Bi0.05O0.65F1.7 phosphors was added.

Text was added.

These tunable emission lights are appropriate for a high color-rendering index to apply phosphor converted UV-light emitting diodes.

Round 2

Reviewer 2 Report

The experiment demonstrated by the authors in supporting materials (Figure S1) only makes sense if both samples are prepared in such a way that they scatter the radiation not absorbed by the sample in the same way. Otherwise, we can generate any emission intensity. Because I believe that the authors did just that (although they do not mention it) I can accept their answer. By the way, in the description of S1 in the second formula, Eu was omitted.